# Microfluidic Electroceuticals Platform for Therapeutic Strategies of Intervertebral Disc Degeneration: Effects of Electrical Stimulation on Human Nucleus Pulposus Cells under Inflammatory Conditions

**DOI:** 10.3390/ijms231710122

**Published:** 2022-09-04

**Authors:** Tae-Won Kim, An-Gi Kim, Kwang-Ho Lee, Min-Ho Hwang, Hyuk Choi

**Affiliations:** 1Department of Medical Sciences, Graduate School of Medicine, Korea University, 148, Gurodong-ro, Guro-gu, Seoul 08308, Korea; 2Division of Mechanical and Biomedical Mechatronics, and Materials Science and Engineering, College of and Engineering, Kangwon National University, 1, Kangwondaehak-gil, Chuncheon-si 24341, Korea

**Keywords:** intervertebral disc degeneration, microfluidics, electrical stimulation, nucleus pulposus, macrophages, inflammation, electroceuticals

## Abstract

The degeneration of an intervertebral disc (IVD) is a major cause of lower back pain. IVD degeneration is characterized by the abnormal expression of inflammatory cytokines and matrix degradation enzymes secreted by IVD cells. In addition, macrophage-mediated inflammation is strongly associated with IVD degeneration. However, the precise pathomechanisms of macrophage-mediated inflammation in IVD are still unknown. In this study, we developed a microfluidic platform integrated with an electrical stimulation (ES) array to investigate macrophage-mediated inflammation in human nucleus pulposus (NP). This platform provides multiple cocultures of different cell types with ES. We observed macrophage-mediated inflammation and considerable migration properties via upregulated expression of interleukin (IL)-6 (*p* < 0.001), IL-8 (*p* < 0.05), matrix metalloproteinase (MMP)-1 (*p* < 0.05), and MMP-3 (*p* < 0.05) in human NP cells cocultured with macrophages. We also confirmed the inhibitory effects of ES at 10 μA due to the production of IL-6 (*p* < 0.05) and IL-8 (*p* < 0.01) under these conditions. Our findings indicate that ES positively affects degenerative inflammation in diverse diseases. Accordingly, the microfluidic electroceutical platform can serve as a degenerative IVD inflammation in vitro model and provide a therapeutic strategy for electroceuticals.

## 1. Introduction

Intervertebral disc (IVD) degeneration is one of the leading causes of low back pain (LBP) and is a growing healthcare problem worldwide. Up to 70% of individuals in the general population experience LBP during their lifetime. Furthermore, LBP results in considerable losses in productivity and increases health care costs [1,2,3].

IVD is an avascular and aneural tissue consisting of a central nucleus pulposus (NP) surrounded by a peripheral annulus fibrosus (AF). The vascular structure and free nerve endings are positioned in the outer third of the AF regions [4]. In healthy IVD, homeostasis is preserved by regulating the balance of diverse molecules, including extracellular matrix (ECM)-modifying enzymes or inflammatory mediators secreted by IVD cells and/or resident immune cells. In contrast, in patients with LBP, IVD tissues exhibit a degenerative microenvironment induced by an imbalance between catabolic and anabolic responses that result in matrix degradation by matrix metalloproteinases (MMPs) and an inflammatory response by various cytokines such as interleukins (ILs) [1,2,3,4]. In addition, neovascularization has been observed in the deep NP regions [5,6]. Multiple lines of evidence have shown that immune cells, including monocytes, neutrophils, macrophages, and T cells, infiltrate the NP regions through these newly formed vascular structures [4,7,8]. Furthermore, marked increases in the expression of various factors associated with degeneration secreted by immune cells or NP cells, including IL-6, IL-8, and MMPs, have been noted in IVD [3,4,9,10]. Thus, an enhanced understanding of the interaction between IVD cells and immune cells could lead to the identification of novel therapeutic targets and establishment of treatment strategies for IVD disease.

In terms of therapeutic strategies, electroceuticals, such as electrical stimulation (ES), have recently been proposed as an effective treatment for a variety of diseases. “Electroceutical” is defined as a technology or tool used to treat diseases by transmitting an electrical signal or stimulus on the body. Studies have shown that ES affects cell migration, wound healing, inflammation, and pain modulation by regulating the expression of molecules [11,12,13,14,15,16,17,18]. More importantly, ES can selectively treat specific targets, including tissues and/or cell populations, whereas conventional chemical drugs can cause side effects in unwanted areas due to their circulation in the body. A study showed that ES significantly diminishes the expression of IL-5 and IL-18 in the hippocampus and IL-6 in the nucleus accumbens in a depression model [19]. Another study demonstrated that capacitively coupled ES induces the upregulation of anabolic matrix molecules, including aggrecan, collagen II, and sulfated glycosaminoglycan, in human NP cells [20]. Our previous study also reported the inhibitory effects of ES on the inflammatory response in human AF cells induced by macrophages by regulating the expression of inflammatory cytokines, such as TNF-α, IL-1β, IL-6, and IL-8 [21,22].

In this study, we developed a microfluidic platform integrated with an electrical stimulation array in which multiple types of cells could be incubated in 3D through the collagen hydrogel. Using this platform, we sought to reproduce the interactions between human NP cells and macrophages that are expected to occur during IVD degeneration. Moreover, we studied the anti-inflammatory effects of ES using this in vitro IVD degeneration model.

## 2. Results

### 2.1. Conditioned Medium from Macrophage THP-1 Cells Induces the Inflammatory Response by Modulating Protein Expression of Inflammatory Mediators and ECM-Modifying Enzymes in Human NP Cells

To investigate the expression pattern of inflammatory mediators and ECM-modifying enzymes in human NP cells treated with soluble factors derived from macrophages, the protein productions of IL-6, IL-8, MMP-1, and MMP-3 were measured in MCM-stimulated human NP cells using ELISA (Figure 1A). Additionally, to verify whether the NF-κB p65 protein acts as a transcriptional factor of catabolic and inflammatory gene encoding translocated into the nucleus in MCM-stimulated NP cells, we analyzed the preferential expression of p65 protein using immunofluorescence.

In this study, to develop an in vitro model of IVD inflammation mediated by macrophages, we first cultured monocyte THP-1 cells with 160 nM PMA for 48 h. Following treatment, the monocyte THP-1 cells were differentiated into macrophages, and macrophage morphology was acquired, which was characterized by the adhesion capacity (Figure 1B).

Second, we examined whether soluble factors derived from differentiated macrophages can induce the activation of NF-κB signaling via translocation of the p65 protein, which initiates detrimental inflammation. Our immunofluorescence images showed the preferential distribution of NF-κB p65 protein to the nucleus in MCM-stimulated NP cells rather than in the cell cytoplasm compared to naïve NP cells. Quantitatively, the average intensity value of the p65 protein increased, and most of the detected intensity was located in the nucleus within 45 min, in contrast to in naïve NP cells (Figure 1C). Additionally, the protein expression of inflammatory mediators and ECM-modifying enzymes, including IL-6, IL-8, MMP-1, and MMP3, was significantly upregulated in MCM-stimulated NP cells compared to in both naïve NP and MCM (Figure 1D). Furthermore, human NP cells exposed to MCM on the microfluidic platform exhibited an identical tendency of protein expression of these target factors compared to those in the culture plate (Figure 1E). To ensure that the results observed after MCM treatment were not due to cytotoxicity, we performed a live/dead assay. Our results showed no differences in cell viability (Figure 1F).

These results demonstrate that soluble factors derived from macrophages can induce an inflammatory response in human NP cells, which is expected to occur during IVD degeneration.

### 2.2. Inflammatory Responses between Human NP and Macrophage THP-1 Cells through 3-Dimensional Migration in Microfluidic Platform

We assessed the migratory properties and protein expression of human NP cells cocultured with macrophage THP-1 to validate the inflammatory response between human NP and macrophages via paracrine signaling and chemotaxis in a microfluidic electrode platform (Figure 2A). In this study, three experimental conditions were evaluated: human NP cocultured with NP (control group 1), macrophages cocultured with macrophages (control group 2), and human NP cells cocultured with macrophages (experimental group).

Fluorescence images showed that both human NP cells and macrophage THP-1 cells exhibited increased migration into the collagen type I hydrogel in the coculture group. In particular, human NP cells exhibited extended filopodia, which are the leading edge of lamellipodia in migrating cells. The migration of human NP cells and macrophages at 48 h was considerably higher than that of the control groups. These features persisted for approximately 72 h of coculture. In contrast, the two control groups did not exhibit any migratory properties on the platform (Figure 2B). This finding implies that there is an interaction between human NP and macrophages through soluble factors derived from each cell.

Furthermore, we examined the protein production of inflammatory mediators and ECM-modifying enzymes in human NP cells and macrophages. Human NP cells cocultured with macrophages expressed significantly higher protein levels of IL-6, IL-8, MMP-1, and MMP-3 compared to naïve NP cells. The macrophages in the coculture group also showed upregulated expression of IL-6 and IL-8 compared to naïve macrophages (Figure 2C). Because the culture of human NP cells or macrophages on fabricated PDMS chips can damage each cell and affect protein production, we assessed cell cytotoxicity using a live/dead assay. Neither human NP cells nor macrophages in any of the groups showed any differences in cell cytotoxicity (Figure 2D).

Together, these results demonstrate that interactions between human NP and macrophages induce the inflammatory milieu in IVD tissues through the secretion of inflammatory factors.

### 2.3. Electrical Stimulation Alleviates the Inflammatory Response Induced by Interaction between Human NP and Macrophages on Microfluidic Electrode Platform

As a therapeutic strategy for IVD degeneration, we investigated the effects of ES on human NP cells and macrophages under inflammatory conditions using a microfluidic electrode platform. ES in various ranges (10, 20, and 50 μA) was conducted on the platform. Human NP cells cocultured with macrophages, applied at 10 μA of ES, displayed significantly attenuated expression of IL-6 and IL-8 compared with human NP cells without ES. In addition, macrophages exposed to 20 μA of ES showed upregulated expression of these factors compared to macrophages without ES. The production of MMP-1 and MMP-3 was not significantly altered in either human NP or macrophages exposed to ES at any of the tested doses (Figure 3A and Table 1).

Because ES can damage cells, we also tested cell cytotoxicity at a dose of 50 μA, which is the maximum dose used in this study. Neither human NP cells nor macrophages with ES showed a difference in cell viability compared to human NP or macrophages in the coculture group (Figure 3B). These findings suggest that ES exerts a positive effect on degenerative IVD by modulating inflammatory mediators.

## 3. Discussion

Our study sought to demonstrate that a potential interplay between human NP and macrophages through paracrine effects could enhance the inflammatory milieu in IVD tissues, exacerbating the cascade of degenerative events. In this study, we implemented our microfluidic coculture platform to reconstitute degenerative IVD models by coculturing human NP cells with macrophages. Using a combined electrode array in this platform, we investigated the effects of ES on the inflammatory response using a therapeutic approach (Figure 4).

Our data showed that under coculture conditions, human NP cells secrete aberrant inflammatory mediators, such as interleukins and matrix-degrading enzymes, leading to macrophage infiltration, matrix breakdown, and cellular damage during IVD degeneration. Intriguingly, considerable migration was observed not only in macrophages, but also in human NP cells. Furthermore, we found that the application of ES under inflammatory conditions significantly reduced the protein production of interleukins such as IL-6 and IL-8, which are believed to be essential for the development of neuropathic pain and chemotaxis of immune cells, by human NP cells in a dose-dependent manner. These findings indicate that ES may be a novel therapeutic tool for the treatment of IVD degeneration.

Immune cell infiltration has been implicated in degenerated IVD. Evidence from some studies showed markedly higher infiltration of macrophages or T cells in degenerated and herniated IVD compared to nondegenerated controls [4,8,23,24]. In addition, one study observed abundant macrophages in the deeper IVD regions of patients with LBP [25]. In support of this observation, several studies have demonstrated the presence of neovascular structures, which are a track for the infiltration of circulating monocytes or neutrophils induced by microenvironmental cues, in degenerated and herniated IVD [5,26,27]. Indeed, in vitro studies have shown that activated macrophages induce the up-regulated inflammatory gene/protein and downregulated ECM-anabolic genes, and attenuated cell proliferation in human NP or AF cells [4,28,29,30,31].

Similarly, we observed that both human NP cells exposed to soluble factors from macrophages and cocultured human NP cells with macrophages expressed considerable amounts of IL-8 and IL-6. IL-8 is a cytokine with a CXC amino acid motif that is involved in chemotaxis activity. It also regulates neovascularization by enhancing the proliferation of endothelial cells and to recruit circulating monocytes or macrophages [3,27,32,33,34]. IL-6 is an inflammatory mediator that can induce hyperalgesia, and higher expression levels have been found in herniated IVD from patients with chronic sciatic pain [4,35,36]. In general, various cytokines and mediators are upregulated via activation of the NF-κB signaling pathway, which is mediated by the nuclear translocation of NF-κB subunits (p65 and p50 proteins) from the cell cytoplasm [4]. In the current study, soluble factors derived from macrophages induced the translocation of the p65 protein into the nucleus, resulting in the upregulation of IL-8, IL-6, and MMPs in human NP cells. In terms of promoting matrix degradation, a marked increase in the production of ECM-modifying enzymes, such as MMPs, has been noted in IVD during degeneration. In particular, the number of immunopositive cells for MMP-1 and MMP-3 increased with the severity of IVD degeneration [37]. MMP-1, a collagenase, predominantly cleaves fibrillary collagens, particularly type 1 collagen. MMP-3 can proteolyze a variety of substrates, including collagen type 2, proteoglycans, and gelatins, which are the major matrix components in NP [4,38]. Similarly, our results revealed that human NP cells cocultured with macrophages upregulated MMP-1 and MMP-3 protein production. Indeed, the migration properties of human NP cells along the 3D collagen hydrogel, which mimics a major ECM component of the outer AF, found in our platform can be mediated by the shifted expression pattern of these enzymes in concert with other cytokines. This observation also supports the hypothesis that interactions between macrophages and human NP cells may contribute not only to the vicious inflammatory microenvironment found in degenerating IVD, but also to NP extrusion in herniated IVD, resulting in nerve root compression and LBP.

In this study, we examined the positive effects of ES in modulating these factors in the presence of an inflammatory response by coculturing human NP cells with macrophages. The human body generates electrical signals for signal transduction, which are controlled by endogenous or exogenous ES. Some studies have demonstrated that ES can accelerate wound healing and attenuate the inflammatory response in diverse tissues by regulating the gene and protein expression of target factors [11,12,14,15,16,18,21,22]. However, the effects of ES on the pathogenesis of degenerated IVD with LBP have not been elucidated. Increasing evidence has shown that ES controls the activation of downstream pathways, such as NF-κB and MAPK signaling, regulating the gene and protein expression of inflammatory mediators [39,40,41,42]. Another study showed that ES reduced hyperalgesia and pain intensity [43]. In addition, ES inhibited spinal extracellular signal-regulated kinase 1/2-cyclooxyhenase-2 pathway activation in rats [44]. Furthermore, in vitro studies have reported that ES in human AF cells drastically decreases inflammatory mediators, such as TNF-a, IL-1b, IL-6, IL-8, and MMP-1 [21,22]. Similarly, our results showed that under coculture conditions, ES at 10 μA significantly reduced the protein production of IL-6 and IL-8 in human NP cells, whereas ES at 20 μA promoted it in macrophages. In terms of how cells and molecules respond to electrical cues, several studies have suggested that ES can activate and perturb the transmembrane electrical potential through the opening of calcium channels or conformational changes in membrane receptors. It is well known that the influx/efflux of calcium ions within intracellular space, especially in the Golgi apparatus influences various cellular responses. In addition, the changes in membrane receptors may modulate the receptor-ligand interactions, resulting in activating the downstream signaling pathways. Consequently, the affinity or specificity of ligand-receptor interactions could be modulated by specific ES parameters including intensity, duration, waveform and frequency [45,46]. Taken together, these results indicate that ES can be a therapeutic tool for the treatment of IVD degeneration by modulating related molecules, and the optimal dose for a specific factor or cell type should be investigated when applying ES to diverse tissues or diseases.

## 4. Materials and Methods

### 4.1. Fabrication of Microfluidic Electrode Platform

The SU-8 photoresists were patterned using standard photolithography on a silicon wafer to fabricate a master mold according to the designed size. A mixture of a polydimethylsiloxane (PDMS SYLGARD 184 A/B, Dow Corning, Midland, MI, USA) base and curing agent in a 9:1 ratio by weight was poured onto the master mold and incubated in a dry oven at 80 °C for 2 h. The PDMS microfluidic chip was bonded to a coverslip using a plasma generator (FEMTO SCIENCE; Hwaseong-si, Gyeonggi-do, KOR). The fabricated devices were autoclaved at 121 °C for 15 min and stored in a dry oven for 2 days to restore hydrophobicity before use. Additionally, we fabricated an electroceutical system to apply electrical stimulation during the coculturing of human NP cells and THP-1 cells. The system was adjustable to apply an electrical stimulation of 10–50 μA. The platform was composed of a microcontroller unit and a six-wire connector that connects six wires supplying alternating current to three microfluidic chips. The internal circuit of the microcontroller has a repetitive process consisting of two different modes and is designed to maintain a constant current using a digital potentiometer as a resistance sensor (MCP4151- 104E/P; Microchip Technology, Chandler, AZ, USA) (Figure 5).

### 4.2. Culture of Primary Human NP Cells

Primary human NP cells were purchased from ScienCell (Carlsbad, CA, USA) and cultured in nucleus pulposus cell medium (NPCM; ScienCell, Carlsbad, CA, USA) containing 2% fetal bovine serum (FBS; ScienCell, Carlsbad, CA, USA), 1% penicillin/streptomycin (P/S; ScienCell, Carlsbad, CA, USA), and nucleus pulposus cell growth supplement (NPCGS; ScienCell, Carlsbad, CA, USA). At approximately 80–90% confluence, the NP cells were subcultured at 5 × 10^5^ cells in poly-D-lysine-coated T-75 cell culture flasks. Human NP cells were used at passage two.

### 4.3. Culture of Immortalized Human Monocyte THP-1 Cells and Production of Conditioned Medium

Human monocyte THP-1 cells were purchased from American Type Culture Collection (ATCC; Manassas, VA, USA) and cultured according to the manufacturer’s instructions. For differentiation into macrophages, monocyte THP-1 cells were cultured in Roswell Park Memorial Institute (RPMI; Gibco-BRL, Gaithersburg, MD, USA) 1640 medium supplemented with 160 nM phorbol myristate acetate (PMA; Gibco-BRL, Gaithersburg, MD, USA), 1% FBS, and 1% P/S. After 48 h, differentiated macrophage THP-1 cells were cultured in Dulbecco’s Modified Eagle Medium Nutrient Mixture F-12 (DMEM/F12; Gibco-BRL, Gaithersburg, MD, USA) containing 1% FBS and 1% P/S for an additional 48 h at a density of 1.0 × 10^6^ cells per flask. The conditioned medium was collected and stored at −80 °C for further experiments. The medium collected from the macrophage THP-1 cells is referred to as macrophage-conditioned medium (MCM).

### 4.4. Generation of Paracrine Signal in the Microfluidic Platform

A collagen type I hydrogel (COL1; BD Bioscience, Santa Clara, CA, USA) was used as an ECM scaffold to generate cell-to-cell paracrine effects in the microfluidic platform. The collagen solution was diluted to a defined concentration (2.0 mg/mL) with 10× PBS, sodium hydroxide (NaOH, 0.5 N), and distilled deionized water. The mixture was then injected into the central collagen channels in the microfluidic chip. For polymerization of the collagen, the devices were placed in a humidified chamber and allowed to gel at 37 °C in a 5% CO_2_ incubator for 30 min. After polymerization, the normal medium was injected into the two side channels. The devices were then placed in an incubator for 1 h to remove any additional bubbles.

### 4.5. Coculturing of Human NP Cells with THP-1 Cells in Microfluidic Platform

First, the monocyte THP-1 cells suspended in DMEM/F12 medium containing 1% FBS and 160 nM PMA were seeded at a concentration of 2.5 × 10^6^ cells/mL in cell culture channels and left to adhere and differentiate for 48 h in an incubator. Non-adherent cells or residual PMA solution were removed by washing with fresh medium. After 48 h, human NP cells were seeded at a concentration of 5.0 × 10^5^ cells/mL into the opposite side channels of the THP-1 channels. The microfluidic chips were kept in an incubator during the experiments, and the medium was changed every two days.

### 4.6. Immunofluorescence Staining

Human NP cells and macrophages cultured on the platform were fixed with 4% paraformaldehyde for 15 min and permeabilized with 0.2% Triton X-100 in phosphate-buffered saline (PBS Gibco-BRL, Gaithersburg, MD, USA) for 5 min. Blocking was performed using 3% bovine serum albumin in PBS (1% *w*/*v*) for 2 h at room temperature. Cells were then incubated with Alexa 488-conjugated phalloidin (Invitrogen, Carlsbad, CA, USA) for 2 h. Subsequently, the cells were stained with 4′,6-diamidino-2- phenylindole (Santa Cruz Biotechnology, Dallas, TX, USA) for 15 min. Images were captured using a confocal microscope (LMS900, ZEISS Inc., Oberkochen, Germany).

### 4.7. Enzyme-Linked Immunosorbent Assay (ELISA)

The protein levels of IL-6, IL-8, MMP-1 and MMP3 in the collected medium were measured using ELISA kits (R&D Systems, Minneapolis, MN, USA), following the manufacturer’s instructions.

### 4.8. Cell Cytotoxicity and Lactate Dehydrogenase Assay

The viability of human NP cells and macrophages cultured on the platform was confirmed using a live/dead assay kit (L3224, Invitrogen, Carlsbad, CA, USA) under the manufacturer’s instructions. Live cells were labelled with green fluorescence (calcein-AM), whereas dead cells were labelled with red fluorescence (ethidium homodimer-1). Fluorescence images were acquired using an EVOS FL auto-cell imaging system (Thermo Fisher Scientific Inc., Waltham, MA, USA).

### 4.9. Statistical Analysis

Data are expressed as the mean ± SEM of three technical replicates and six individual experiments using independent cell cultures. One-way analysis of variance and Bonferroni’s correction post-hoc test were used to assess the differences among the experimental groups. All statistical analyses were performed using GraphPad Prism 9.0 (GraphPad Software, La Jolla, CA, USA). Statistical significance was expressed as * *p* < 0.05, ** *p* < 0.01, *** *p* < 0.001, ^#^ *p* < 0.05, and ^##^ *p* < 0.01.

## 5. Conclusions

We propose an interaction between human NP and macrophages in an in vitro model and examine the effects of ES in degenerative IVD conditions using a microfluidic electroceutical platform. This can mimic the immune response expected to occur during the development of IVD degeneration in the human body. Our results suggest that the biological interactions between human NP cells and infiltrating macrophages lead to increasingly severe inflammatory conditions, and a better understanding of these interactions provides therapeutic target molecules of IVD degeneration for the application of ES. In addition, our platform provides a reliable system to study pathomechanisms in diverse diseases induced by cell-to-cell interplay and to assess the effects of ES in a 3D microenvironment.

## Figures and Tables

**Figure 1 ijms-23-10122-f001:**
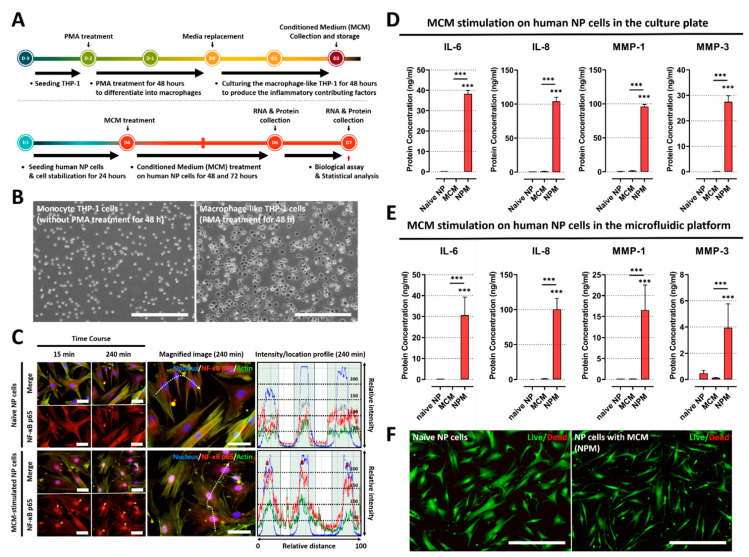
Effects of soluble factors on the culture medium of macrophage THP-1 cells on human NP cells. (**A**) Schematic of experimental timeline for MCM production of and paracrine effect replication on human NP cells stimulated by MCM. (**B**) The images show that human monocyte THP-1 cells differentiate into activated THP-1 macrophages cells upon PMA treatment for 48 h (scale bar = 400 μm). (**C**) MCM stimulation on human NP cells for degenerative IVD conditions. Fluorescence image and quantification of fluorescence intensity of NF-κB p65 in human NP cells stimulated by MCM are shown. The NF-κB p65 protein is preferentially distributed in the nucleus rather than the cytoplasm (red arrow). This can induce degenerative conditions in human NP cells as the p65 protein acts as a transcription factor (scale bar = 125 μm). (**D**) The results show that MCM-exposed human NP cells produce inflammatory mediators and ECM-modifying enzymes compared to naïve NP cells or MCM, in both conventional culture plates and (**E**) our microfluidic platform. (**F**) Assessment of human NP cytotoxicity with/without MCM using a live/dead assay. MCM-cultured human NP cells did not exhibit any difference in cell cytotoxicity compared with naïve NP cells (scale bar = 400 μm). *** *p* < 0.001 as compared with naïve NP cells. The line indicates comparison with each group.

**Figure 2 ijms-23-10122-f002:**
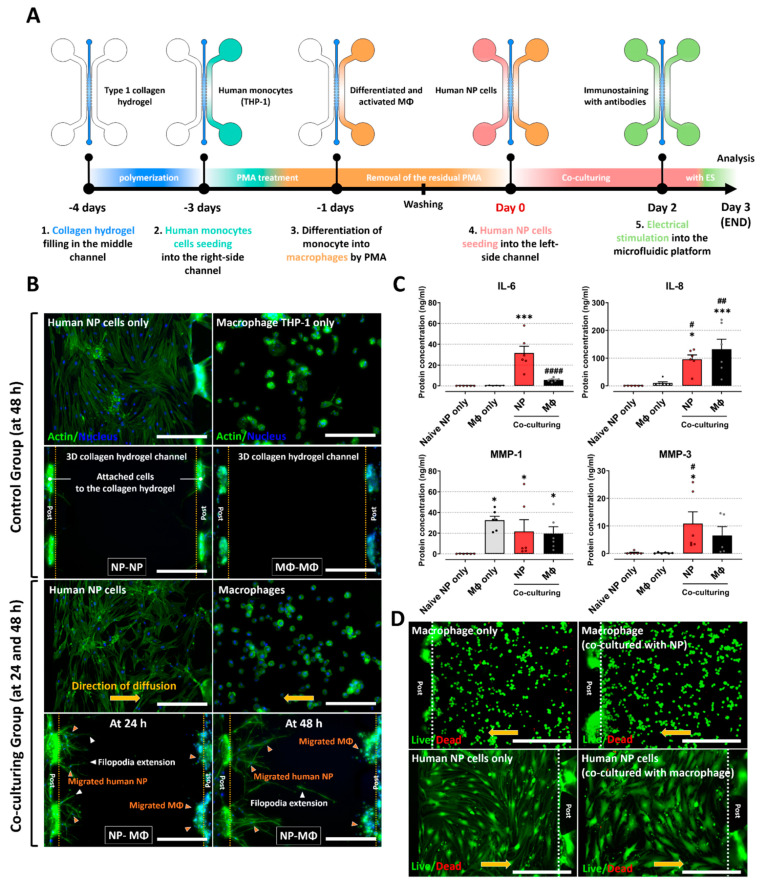
Interaction between human NP cells and macrophage THP-1 cells on the microfluidic platform. (**A**) Schematic of experimental setup and timeline on the microfluidic platform. (**B**) Significant migration of human NP cells and macrophage THP−1 cells (NP−MΦ group) is observed in the central collagen hydrogel channels at culture times of 24 and 48 h (orange arrowhead), whereas the control groups (NP−NP or MΦ−MΦ) did not exhibit any difference in the migration properties at a culture time of 48 h. In addition, the human NP cells show directional extension of filopodia (white arrowhead), which are the leading edge of lamellipodia in migrating cells, from human NP cells toward the macrophage channel (scale bar = 400 μm). (**C**) Protein production in human NP cells and macrophages on the microfluidic platform. The coculturing groups show higher protein production due to inflammatory mediators and ECM-modifying enzymes than naïve NP cells or naïve macrophages. (**D**) Assessment of human NP or macrophage cytotoxicity on the microfluidic platform using a live/dead assay. Both human NP cells and macrophages did not show any difference in cell cytotoxicity compared with naïve NP cells or naïve macrophages (scale bar = 400 μm). * *p* < 0.05, and *** *p* < 0.001 as compared to naïve NP cells. ^#^
*p* < 0.05, ^##^ *p* < 0.01, and ^####^ *p* < 0.0001 as compared to naïve macrophages. Each dot represents the mean of six samples in an independent experiment.

**Figure 3 ijms-23-10122-f003:**
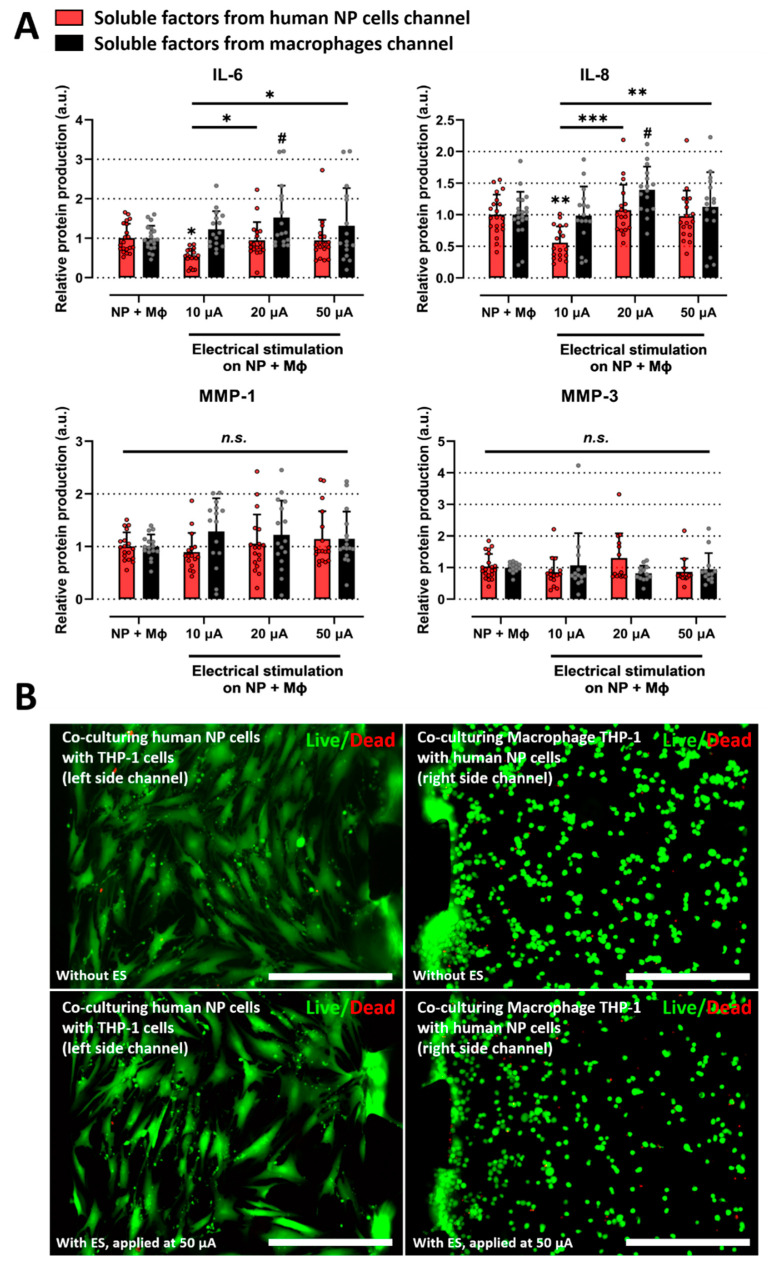
Effects of electrical stimulation on the microfluidic platform under coculturing conditions. (**A**) Protein production in human NP cells and macrophages with/without ES on the microfluidic platform. The ES, applied at 10 μA, inhibits the protein production of inflammatory mediators such as IL-6 and IL-8 in degenerative human NP cells. (**B**) Assessment of human NP or macrophage cytotoxicity with/without ES applied at 50 μA on the microfluidic platform using a live/dead assay. 50 μA is the maximum dose used in this study. The cells did not show a difference in cell cytotoxicity (scale bar = 400 μm). * *p* < 0.05, ** *p* < 0.01, *** *p* < 0.001, and n.s., no significant difference, as compared with human NP cells cocultured with macrophages. ^#^ *p* < 0.05 as compared with macrophages cocultured with human NP cells. The line indicates the comparison with each group. Each dot represents the mean of six samples in an independent experiment.

**Figure 4 ijms-23-10122-f004:**
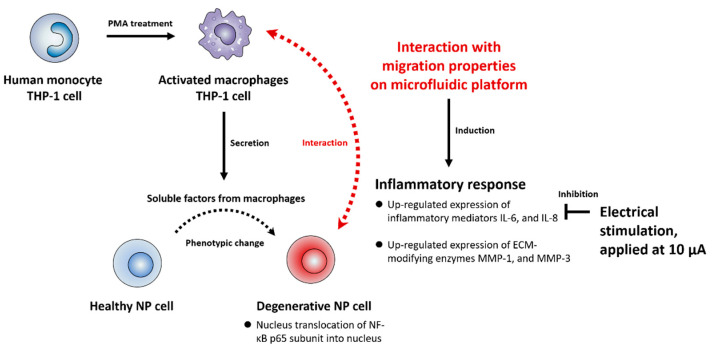
Schematic summary of the interaction between human NP and macrophages in the in vitro model and effects of ES on the degenerative IVD models. In this study, we first confirmed that soluble factors derived from macrophages activate the NF-κB signaling pathway in human NP cells, inducing the protein production of IL-6, IL-8, MMP-1, and MMP-3 via translocation of NF-κB p65 protein into the nucleus. Second, we also mimicked a degenerative IVD in vitro model by coculturing human NP cells and macrophages by using a newly developed microfluidic platform. The interactions between the human NP cells and macrophages induced inflammatory milieu through the production of inflammatory mediators and ECM-modifying enzymes. Third, ES demonstrated an inhibitory effect of the production of inflammatory mediators IL-6 and IL-8 in human NP cells under these conditions. Cumulatively, these findings can provide therapeutic target molecules for macrophage-mediated inflammation in degenerative IVD.

**Figure 5 ijms-23-10122-f005:**
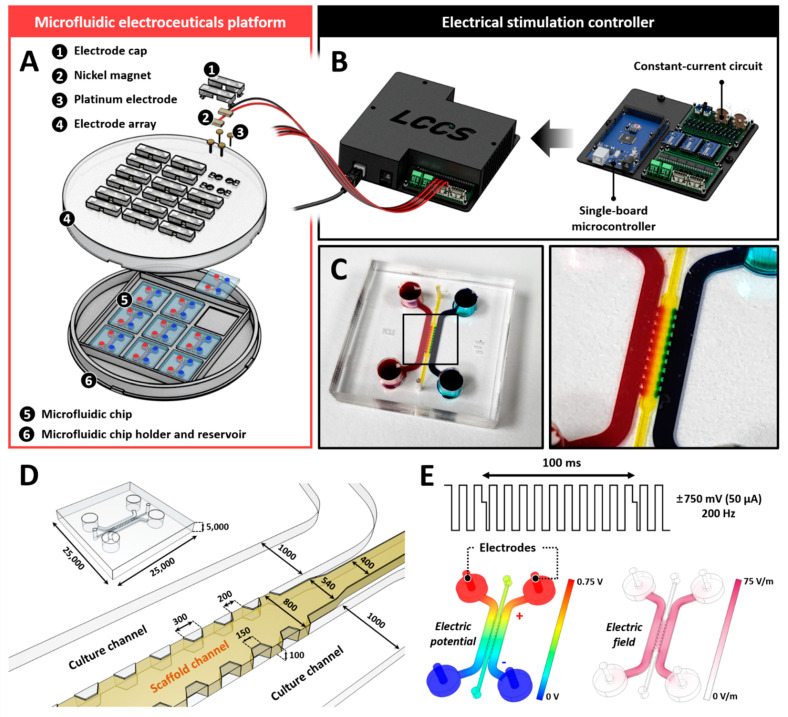
Microfluidic electroceuticals platform. (**A**) The microfluidic electroceuticals platform comprises two major units: an electrical stimulation controller and a microfluidic chip array. (**B**) The electrical stimulation controller regulates the micro-current strength. The output microcurrent strength in the microfluidic platform is in the range of 0−50 μA. The output timing of the current can be adjusted using the microcontroller. The internal circuit, combined with the controller, was developed to maintain a constant current during the cell culture process. (**C**) The microfluidic chips were fabricated using standard lithography. They comprise of two distinct cell culture chambers connected through a collagen type 1 hydrogel channel for various reagents or soluble factors derived from cells to diffuse. (**D**) Schematic and extended view of the microfluidic chips. (**E**) The electric field within the channel was applied at the biphasic signals (±750 mV, 50 µA, 200 Hz), which constitute the maximum dose of electrical stimulation used in this study. Simulation images show electric potential formation or field differences. All scales are represented in μm.

**Table 1 ijms-23-10122-t001:** Protein concentrations of inflammatory cytokines and ECM-modifying enzymes in NP cells and macrophages.

Factor (ng/mL)	Naïve Human NP	Human NP Cells Cocultured with Macrophages			
		without ES	+ ES, 10 μA	+ ES, 20 μA	+ ES, 50 μA
IL-6	0.02 ± 0.02	31.49 ± 6.58 ^###^	18.15 ± 6.21 *	27.83 ± 5.67	28.54 ± 7.14
IL-8	0.30 ± 0.14	96.01 ± 15.45 ^#^	54.12 ± 14.83 **	96.11 ± 13.93	89.65 ± 18.38
MMP-1	0.13 ± 0.13	21.67 ± 11.40 ^#^	17.69 ± 9.05	15.67 ± 6.55	18.05 ± 8.48
MMP-3	0.23 ± 0.49	10.24 ± 2.07 ^#^	8.76 ± 1.83	11.28 ± 1.91	8.62 ± 2.0
**Factor** **(ng/mL)**	**naïve macrophage**	**Macrophages cocultured with human NP cells**			
		**without ES**	**+ ES, 10 μA**	**+ ES, 20 μA**	**+ ES, 50 μA**
IL-6	0.20 ± 0.23	5.42 ± 1.08 ^####^	6.33 ± 0.024	7.96 ± 0.49 *	6.26 ± 0.20
IL-8	10.17 ± 11.53	143.94 ± 52.95 ^##^	106.50 ± 44.01	170.55 ± 26.82 *	118.14 ± 46.39
MMP-1	32.53 ± 9.58	15.35 ± 7.85	23.54 ± 12.94	23.81 ± 11.63	21.56 ± 8.05
MMP-3	0.18 ± 0.22	6.57 ± 1.55	4.72 ± 1.03	5.01 ± 1.15	5.69 ± 1.39

^#^ *p* < 0.05, ^##^ *p* < 0.01, ^###^ *p* < 0.001, and ^####^ *p* < 0.0001 compared to naïve human NP cells or macrophages, respectively. * *p* < 0.05 and ** *p* < 0.01 compared to cocultured human NP cells or macrophages without ES group, respectively.

## Data Availability

The datasets generated and/or analyzed during the current study are available from the corresponding author upon reasonable request.

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
