# Peer review of "Microfluidic Electroceuticals Platform for Therapeutic Strategies of Intervertebral Disc Degeneration: Effects of Electrical Stimulation on Human Nucleus Pulposus Cells under Inflammatory Conditions"

_ijms, 2022, doi:10.3390/ijms231710122_

Round 1

Reviewer 1 Report

The manuscript by Kim et al study the interactions between human NP cells and macrophages that are expected to occur during IVD degeneration. The authors have developed a microfluidic platform integrated with an electrical stimulation array, in which multiple types of cells could be incubated in 3D through the collagen hydrogel.

It is stated in the abstract that the macrophage-mediated inflammation is strongly associated with IVD degeneration. However, there is no related reference and discussion in the manuscript except for not-mentioned ref 26. Moreover it is unclear how the study setting is related IVD degeneration and degenerative inflammation in diverse diseases.

Macrophage-conditioned medium (MCM) and soluble factors derived from macrophages are not synonymous. The effects observed following treatment of MCM can be dependent upon soluble factors derived from macrophages, but can be not. In the current study there is no experimental evidence confirming that there are identified soluble factors in the system that are derived from macrophages.

It is strongly claimed in the whole manuscript that and current study by menas of microfluidic electroceuticals platform can lead to identification of novel therapeutic targets and establishment of treatment strategies for IVD disease. However, I find no information on IVD degeneration model in this paper.

Author Response

Microfluidic Electroceuticals Platform for Therapeutic Strategies of Intervertebral Disc Degeneration: Effects of electrical stimulation on human nucleus pulposus cells under inflammatory conditions

Tae-Won Kim 1,, An-Gi Kim 1,, Kwang-Ho Lee 2, Min-Ho Hwang 1,* and Hyuk Choi 1,*

Reviewer 1

Comments and Suggestions for Authors

The manuscript by Kim et al study the interactions between human NP cells and macrophages that are expected to occur during IVD degeneration. The authors have developed a microfluidic platform integrated with an electrical stimulation array, in which multiple types of cells could be incubated in 3D through the collagen hydrogel.

It is stated in the abstract that the macrophage-mediated inflammation is strongly associated with IVD degeneration. However, there is no related reference and discussion in the manuscript except for not-mentioned ref 26.

  • We thank you for your suggestion. We have added related reference in the section of Discussion.

4.(Manuscript Ref. No.) Risbud, M. V.; Shapiro, I. M., Role of cytokines in intervertebral disc degeneration: pain and disc content. Nat Rev Rheumatol 2014, 10, (1), 44-56.

  1. Kim, J. H.; Studer, R. K.; Sowa, G. A.; Vo, N. V.; Kang, J. D., Activated macrophage-like THP-1 cells modulate anulus fibrosus cell production of inflammatory mediators in response to cytokines. Spine (Phila Pa 1976) 2008, 33, (21), 2253-9.
  2. Li, X.-C.; Luo, S.-J.; Fan, W.; Zhou, T.-L.; Tan, D.-Q.; Tan, R.-X.; Xian, Q.-Z.; Li, J.; Huang, C.-M.; Wang, M.-S., Macrophage polarization regulates intervertebral disc degeneration by modulating cell proliferation, inflammation mediator secretion, and extracellular matrix metabolism. Frontiers in Immunology 2022, 13.
  3. Nakazawa, K. R.; Walter, B. A.; Laudier, D. M.; Krishnamoorthy, D.; Mosley, G. E.; Spiller, K. L.; Iatridis, J. C., Accumulation and localization of macrophage phenotypes with human intervertebral disc degeneration. The Spine Journal 2018, 18, (2), 343-356.
  4. Yang, C.; Cao, P.; Gao, Y.; Wu, M.; Lin, Y.; Tian, Y.; Yuan, W., Differential expression of p38 MAPK α, β, γ, δ isoforms in nucleus pulposus modulates macrophage polarization in intervertebral disc degeneration. Scientific Reports 2016, 6, (1), 22182.

Revised Manuscript

Page 12

Discussion

Immune cell infiltration has been implicated in degenerated IVD. Evidence from some studies showed markedly higher infiltration of macrophages or T-cells in degenerated and herniated IVD compared with nondegenerated controls [3, 10, 25, 26]. In addition, one study observed abundant macrophages in the deeper IVD regions of patients with LBP [27]. In support of this observation, several studies have demonstrated the presence of neovascular structures, which are a track for the infiltration of circulating monocytes or neutrophils induced by microenvironmental cues, in degenerated and herniated IVD [7, 28, 29]. Indeed, in vitro studies have shown that activated macrophages induce the upregulated inflammatory gene/protein and downregulated ECM-anabolic genes, and attenuate cell proliferation in human NP or AF cells [4, 28-31].

Moreover, it is unclear how the study setting is related IVD degeneration and degenerative inflammation in diverse diseases.

  • We thank you for your suggestion. Previous research has demonstrated that proteins released by infiltrating macrophages and macrophages induce IVD inflammation and degeneration (Ref. 4, 28-31). In this study, the monocyte THP-1 cells were treated with PMA to induce their differentiation into macrophages that release pro-inflammatory cytokines and inflammatory mediators. Thus, IVD degeneration conditions were generated by co-culturing macrophages with human NP cells. In this study, co-cultured human NP cells exhibited a dramatic increase in ECM-catabolic enzymes and inflammatory mediators (Figure 3C).

Supporting Information

Page 12

Materials and Methods

2.5. Coculturing of human NP cells with THP-1 cells in microfluidic platform

First, the monocyte THP-1 cells suspended in DMEM/F12 medium containing 1% FBS and 160 nM PMA were seeded at a concentration of 2.5  106 cells/mL in cell culture channels and left to adhere and differentiate for 48 h in an incubator. Non-adherent cells or residual PMA solution were removed by washing with fresh medium. After 48 h, human NP cells were seeded at a concentration of 5.0  105 cells/mL into the opposite side channels of the THP-1 channels. The microfluidic chips were kept in an incubator during the experiments, and the medium was changed every two days.

Macrophage-conditioned medium (MCM) and soluble factors derived from macrophages are not synonymous. The effects observed following treatment of MCM can be dependent upon soluble factors derived from macrophages, but can be not. In the current study there is no experimental evidence confirming that there are identified soluble factors in the system that are derived from macrophages.

  • Thank you for pointing this out. In our previous study, we confirmed that IL-1beta, TNF-alpha, IL-6, and IL-8 were the major pro-inflammatory proteins for inflammatory and degenerative conditions in IVD between many macrophage-secreted proteins. Also, when these proteins were added to human disc cells (AF and NP), the expression of degenerative ECM-modifying enzymes (MMPs and TIMPs), collagen, and inflammatory mediators (IL-6 and IL-8) proteins and their genes went up in a way that was statistically significant (Attached Ref.1-5).
  • In addition, earlier research has demonstrated that chemicals produced by macrophages can induce inflammation and degenerative conditions in IVD cells. (Attached Ref. 6-7)

Our previous studies

  1. Hwang, M. H.; Kim, K. S.; Yoo, C. M.; Shin, J. H.; Nam, H. G.; Jeong, J. S.; Kim, J. H.; Lee, K. H.; Choi, H., Photobiomodulation on human annulus fibrosus cells during the intervertebral disk degeneration: extracellular matrix-modifying enzymes. Lasers Med Sci 2016, 31, (4), 767-77.
  2. Hwang, M. H.; Shin, J. H.; Kim, K. S.; Yoo, C. M.; Jo, G. E.; Kim, J. H.; Choi, H., Low level light therapy modulates inflammatory mediators secreted by human annulus fibrosus cells during intervertebral disc degeneration in vitro. Photochem Photobiol 2015, 91, (2), 403-10.
  3. Hwang, M. H.; Son, H. G.; Lee, J. W.; Yoo, C. M.; Shin, J. H.; Nam, H. G.; Lim, H. J.; Baek, S. M.; Park, J. H.; Kim, J. H.; Choi, H., Photobiomodulation of extracellular matrix enzymes in human nucleus pulposus cells as a potential treatment for intervertebral disk degeneration. Sci Rep 2018, 8, (1), 11654.
  4. Hwang, M. H.; Son, H. G.; Lee, J. W.; Yoo, C. M.; Shin, J. H.; Nam, H. G.; Lim, H. J.; Baek, S. M.; Park, J. H.; Kim, J. H.; Choi, H., Phototherapy suppresses inflammation in human nucleus pulposus cells for intervertebral disc degeneration. Lasers Med Sci 2018, 33, (5), 1055-1064.
  5. Shin, J.; Hwang, M.; Back, S.; Nam, H.; Yoo, C.; Park, J.; Son, H.; Lee, J.; Lim, H.; Lee, K.; Moon, H.; Kim, J.; Cho, H.; Choi, H., Electrical impulse effects on degenerative human annulus fibrosus model to reduce disc pain using micro-electrical impulse-on-a-chip. Sci Rep 2019, 9, (1), 5827.

Other group’s studies

  1. Kim, J. H.; Studer, R. K.; Sowa, G. A.; Vo, N. V.; Kang, J. D., Activated macrophage-like THP-1 cells modulate anulus fibrosus cell production of inflammatory mediators in response to cytokines. Spine (Phila Pa 1976) 2008, 33, (21), 2253-9.
  2. Li, X.-C.; Luo, S.-J.; Fan, W.; Zhou, T.-L.; Tan, D.-Q.; Tan, R.-X.; Xian, Q.-Z.; Li, J.; Huang, C.-M.; Wang, M.-S., Macrophage polarization regulates intervertebral disc degeneration by modulating cell proliferation, inflammation mediator secretion, and extracellular matrix metabolism. Frontiers in Immunology 2022, 13.

It is strongly claimed in the whole manuscript that and current study by menas of microfluidic electroceuticals platform can lead to identification of novel therapeutic targets and establishment of treatment strategies for IVD disease. However, I find no information on IVD degeneration model in this paper.

  • We thank you for your sincere review. Based on what we said above and what we wrote in our manuscript, this study confirmed and verified changes in expression of developmental ECM-modifying enzymes and inflammatory mediators in cells through MCM treatment or Macrophages-Human NP cells co-culture as an IVD degeneration model.

Reviewer 2 Report

Microfluidic Electroceuticals Platform for Therapeutic Strategies of Intervertebral Disc Degeneration: Effects of electrical stimulation on human nucleus pulposus cells under inflammatory conditions

The authors present a microfluidics platform in which they study interactions between human NP cells and macrophages that are expected to occur during IVD degeneration. The cells are able to interact via a separating collagen barrier through which molecules of interest can diffuse and through which cells can migrate. The platform also allowed for the application of electrical fields to the cells and the anti-inflammatory effects of the applied electric fields was studied.

General Comments:

The experimental method seems well conceived and the authors make a very clear argument for the relevance of the study and its results. The manuscript is excellently written, with a structure and language that is easy to follow throughout. While I am not an expert on the specific biological questions that are approached in the work, the motivations for the experimental steps, their descriptions and the analysis of results are thorough enough to follow and seem valid. I would highly recommend the publication of the manuscript after some minor revisions.

Specific comments:

1.      In figure 1 I would like to see a schematic that shows the position of the electrodes in relation to the channels. Maybe this is shown in E? If so then it would help the reader if they were clearly labelled. In C it is easy to see the diffusion of the dyes into the collagen channel. It would be nice to point this out.

2.      In figure 2 the text size in D and E could be increased. Also, the blue/red/green text in C is extremely difficult to read. In the figure text it refers to red arrows but these appear to be orange (as in figure 3 B).

3.      Figure 3 C. Text here is also very small.

4.      It would be very useful to know the approximate diffusion times of the various molecules through the collagen channel. If this is long compared to the culture times then it could be important. For example, do the concentrations become equilibrated over the experimental time scale or are there gradients and for how long do they persist?

5.      It would be good for the authors to motivate the choices of ±750 mV, 200Hz and 10-50mA. How do these values relate to those that are possible in a potential ES therapeutic intervention?

6.      Lastly, how are the various proteins of interest affected by the applied electric field? The manuscript would be more thorough with a comment on the effect/lack of effect of the electric field on the collection of proteins for the culture channels.

Author Response

Microfluidic Electroceuticals Platform for Therapeutic Strategies of Intervertebral Disc Degeneration: Effects of electrical stimulation on human nucleus pulposus cells under inflammatory conditions

Tae-Won Kim 1,, An-Gi Kim 1,, Kwang-Ho Lee 2, Min-Ho Hwang 1,* and Hyuk Choi 1,*

Reviewer 2

Comments and Suggestions for Authors

Microfluidic Electroceuticals Platform for Therapeutic Strategies of Intervertebral Disc Degeneration: Effects of electrical stimulation on human nucleus pulposus cells under inflammatory conditions

The authors present a microfluidics platform in which they study interactions between human NP cells and macrophages that are expected to occur during IVD degeneration. The cells are able to interact via a separating collagen barrier through which molecules of interest can diffuse and through which cells can migrate. The platform also allowed for the application of electrical fields to the cells and the anti-inflammatory effects of the applied electric fields was studied.

General Comments:

The experimental method seems well conceived and the authors make a very clear argument for the relevance of the study and its results. The manuscript is excellently written, with a structure and language that is easy to follow throughout. While I am not an expert on the specific biological questions that are approached in the work, the motivations for the experimental steps, their descriptions and the analysis of results are thorough enough to follow and seem valid. I would highly recommend the publication of the manuscript after some minor revisions.

Specific comments:

  1. In figure 1 I would like to see a schematic that shows the position of the electrodes in relation to the channels. Maybe this is shown in E? If so then it would help the reader if they were clearly labelled. In C it is easy to see the diffusion of the dyes into the collagen channel. It would be nice to point this out. 2.In figure 2 the text size in D and E could be increased. Also, the blue/red/green text in C is extremely difficult to read. In the figure text it refers to red arrows but these appear to be orange (as in figure 3 B). 3. Figure 3 C. Text here is also very small.
  • We thank you for your suggestion and sincere review. We have now revised our figures accordingly as your suggestion.
  1. It would be very useful to know the approximate diffusion times of the various molecules through the collagen channel. If this is long compared to the culture times then it could be important. For example, do the concentrations become equilibrated over the experimental time scale or are there gradients and for how long do they persist?
  • In this work, the macrophages treated with PMA secrete pro-inflammatory cytokines (MW approximately 5-40 kDa) lasting 48–72 hours as de novo synthetics. Therefore, it takes around 12 hours for proteins to reach the NP cell channel on the opposite side via the collagen channel, and this occurrence can extend for up to 96 hours.
  1. It would be good for the authors to motivate the choices of ±750 mV, 200Hz and 10-50mA. How do these values relate to those that are possible in a potential ES therapeutic intervention?
  • Thank you for pointing this out. The parameters used in this study may not be practically applicable in clinics, immediately. Since ES requires to have the target tissues or cells with sufficient energy delivered, and the optimal dose may be the first requirement for clinical application. In addition, fusion of ES with delivery system may be suggested as a strategy for clinical practice. This in vitro model represents one of the approaches that can be explored in future research.
  1. Lastly, how are the various proteins of interest affected by the applied electric field? The manuscript would be more thorough with a comment on the effect/lack of effect of the electric field on the collection of proteins for the culture channels.
  • We thank you for your suggestion. We have added more information in the section of Discussion.

Revised Manuscript

Page 12

Discussion

In this study, we examined the positive effects of ES in modulating these factors in the presence of an inflammatory response by coculturing human NP cells with macrophages.

Similarly, our results showed that under co-culture conditions, ES at 10 μA significantly reduced the protein production of IL-6 and IL-8 in human NP cells, whereas ES at 20 μA promoted it in macrophages. In terms of how cells and molecules respond to electrical cues, several studies have suggested that ES can activate and perturb the transmembrane electrical potential through the opening of calcium channels or conformational changes in membrane receptors. It is well known that the influx/efflux of calcium ions within intracellular space, especially in the Golgi apparatus influences various cellular responses. In addition, the changes in membrane receptors may modulate the receptor-ligand interactions, resulting in activating the downstream signaling pathways. Consequently, the affinity or specificity of ligand-receptor interactions could be modulated by specific ES parameters including intensity, duration, waveform and frequency [45, 46]. Taken together, these results indicate that ES can be a therapeutic tool for the treatment of IVD degeneration by modulating related molecules, and the optimal dose for a specific factor or cell type should be investigated when applying ES to diverse tissues or diseases.

Round 2

Reviewer 1 Report

I thank author's response and discussion. The paper should be accepted for publication now.